# Mechanotransduction in the Cardiovascular System: From Developmental Origins to Homeostasis and Pathology

**DOI:** 10.3390/cells8121607

**Published:** 2019-12-11

**Authors:** Gloria Garoffolo, Maurizio Pesce

**Affiliations:** 1Unità di Ingegneria Tissutale Cardiovascolare, Centro Cardiologico Monzino, IRCCS, Via Parea, 4, I-20138 Milan, Italy; maurizio.pesce@ccfm.it; 2PhD Program in Translational and Molecular Medicine DIMET, Università di Milano - Bicocca, 20126 Milan, Italy

**Keywords:** mechanotransduction, YAP/TAZ, stiffness, fibrosis, stromal cells

## Abstract

With the term ‘mechanotransduction’, it is intended the ability of cells to sense and respond to mechanical forces by activating intracellular signal transduction pathways and the relative phenotypic adaptation. While a known role of mechanical stimuli has been acknowledged for developmental biology processes and morphogenesis in various organs, the response of cells to mechanical cues is now also emerging as a major pathophysiology determinant. Cells of the cardiovascular system are typically exposed to a variety of mechanical stimuli ranging from compression to strain and flow (shear) stress. In addition, these cells can also translate subtle changes in biophysical characteristics of the surrounding matrix, such as the stiffness, into intracellular activation cascades with consequent evolution toward pro-inflammatory/pro-fibrotic phenotypes. Since cellular mechanotransduction has a potential readout on long-lasting modifications of the chromatin, exposure of the cells to mechanically altered environments may have similar persisting consequences to those of metabolic dysfunctions or chronic inflammation. In the present review, we highlight the roles of mechanical forces on the control of cardiovascular formation during embryogenesis, and in the development and pathogenesis of the cardiovascular system.

## 1. The Critical Role of Mechanical Forces in Development, Physiology and Pathology

All cells and tissues in multicellular organisms are continuously subjected to mechanical stresses. These forces have various origins, from pressure forces linked to gravity, to forces related to motion (e.g., muscle contraction, blood flow). They are therefore part of various morphogenetic events. It is now accepted that these applied forces are able to modify cellular behavior by affecting transcriptional cell machinery and driving the cell fate specification and differentiation [1]. Various genes and signaling molecules involved in tissue patterning, organogenesis and regeneration have been discovered [2]; however, it is still not well characterized how these signals are distributed in the 3D environment to induce the formation of living tissue with highly specialized shapes and unique functions.

Physical cues were postulated to play a role in tissue development over a century ago on the basis that changes in the three-dimensional shape of any structure, including living cells and tissues, are consequences of the action of a force acting on a mass [3]. Mechanical forces regulate the shaping of the living embryo from the earliest developmental stages following egg activation, until early asymmetric cellular divisions and establishment of initial embryonic polarity. Thereafter, the mechanotransduction process continues in adult life, contributing to tissue growth, homeostasis and, finally, disease programming.

In the present contribution, we will focus on the role of mechanical forces in controlling the morphogenesis of the cardiovascular system and on their new emerging role as a pathology determinant. We will also describe how traction forces exerted locally by single cells or forces (e.g., laminar/perturbed shear stress, constant/oscillatory pressure) propagated passively in the tissues are converted into signals regulating intracellular biochemistry and gene expression underlying pathology progression.

### Definition of Cell Mechanotransduction: Outside-In and Inside-Out Communication in the Complex 3D Environment

Similar to classical ligand/receptor interactions, mechanotransduction requires binding of cell surface receptors to their ligands immobilized into the extracellular matrix (ECM), or expressed at the surface of adjacent cells. Differences in the mechanical features of the ECM, or in the geometrical arrangement of receptor binding motifs, can have a direct readout on cell proliferation, differentiation and migratory responses. Mechanical cues are converted into biochemical signals by activation of intracellular cascades transmitted via the cytoskeleton and their components, for example the acto-myosin ‘stress’ fibers [4], the microtubules [5], the scaffolding proteins [6], and various kinases and phosphatases [7] (Figure 1).

Cell-generated or extrinsic mechanical forces are able to modify global gene expression through mechanoresponsive transcription factors in a receptor-independent manner. The most notable mechanically-regulated transcription co-regulators, YAP/TAZ (components of the so called ‘Hippo’ pathway) [8] or MRTF-SRF [9], require active Rho-GTPase signaling and actomyosin-mediated contractility to promote translocation of these transcription factors from the cytoplasm to the nucleus, where they initiate their transcriptional function [10]. The impact of tissue mechanics on activation of cell mechanosensing is reflected by their elasticity described by the so called ‘elastic (Young’s) modulus’. Due to their specific extracellular matrix composition, each tissue and organ has a reference elastic modulus (measured in Pascal, Pa), which is finely tuned in relationship to their specific functions. Bone is the stiffest tissue in the body for its primary function to provide skeleton structure and mechanical support, while mechanically static tissues, such as the brain, exhibit the lowest stiffness values (Figure 2) [11,12]. At a cellular level, tissue matrix elasticity has been found to directly correlate with differentiation of cells with progenitor characteristics. For example, mesenchymal progenitors upregulate neurogenic or osteogenic transcripts onto compliant (0.1–1 kPa) and stiff matrices (≥34 kPa), respectively [13].

Emerging evidence shows that, in addition to activating mechanical-dependent transcriptional pathways, mechanical stress might also alter nuclear architecture, and modify the general activation of the chromatin through a direct connection of the nuclear lamina with the contractile cytoskeleton network [14]. For example, it was reported that inhibition of stress fiber contractility by actin depolymerization significantly decreases strain transfer to the nucleus and this attenuates its deformation with consequent reduction of mechano-dependent transcription factors translocation through the nuclear pores [14,15] (Figure 1). Generation of cell forces may not only lead to dynamic deformation of the nucleus and trafficking in and out of transcription factors, but may also alter the nuclear topology resulting into different activation states [15] by dislodging chromatin from the nuclear periphery and inducing a relocation of gene loci within the nuclear interior, facilitating gene transcription [16] (Figure 1). In addition, cell forces may even contribute to activate genes located in silent heterochromatin domains by promoting access for transcription machinery, thus re-directing the cell fate towards specific phenotypes [17]. For example, forces generated by the cytoskeleton are propagated to nuclear lamina to physically stretch the chromatin, and facilitating the binding of the RNA Polymerase II with the transcription factors [18]. Conversely, forcing cells to acquire specific geometries onto micropatterned surfaces or to align along specific directions onto microgrooved adhesion patterns, determines substantial changes in epigenetic modifications that may reflect in global differences in cell programming and differentiation [19].

The importance of mechanical signaling is especially relevant for the cardiovascular system. The myocardium is indeed continuously contracting and this occurs for over three billion times during the course of the average lifetime. Proper functioning of cardiac, valve and vascular tissues is dependent on their active and passive mechanical properties, and alteration of these characteristics is believed to determine a pathological evolution of tissues.

## 2. Mechanical Forces in Development and Pathology of the Myocardium

### 2.1. Mechanical Regulation of Cardiac Development

The heart is the first functional organ to form in the embryo, ensuring a supply of oxygen and nutrients to the developing tissues. Changes in pressure, strain and wall shear stress are involved in cardiac morphogenesis toward a complete multi-chambered structure with associated fibrous valves, starting from a linear valve-less tube [20]. A first example of mechanical regulation of cardiogenesis is the heart tube formation in avian embryos [21]. In this model, early cardiac progenitor cells reside in the lateral plate mesoderm [21] but maintain close contacts with the underlying endoderm [22], forming the so-called cardiac crescent [23]. These epithelia move towards the midline and then fuse above the anterior intestinal portal to form the heart tube [24]. This implicates an active mechanical role for the endoderm during this process. In particular, it serves as a mechanical substrate for cardiac mesoderm migration, which drives cell cytoskeletal contractions within the endoderm [25]. Supporting this hypothesis, inhibition of endodermal shortening, using myosin-II inhibitor, leads to cardia bifida and abnormal cardiac morphogenesis [26]. Modifications in mechanical compliance of ECM in endoderm is necessary to address cells toward their appropriate positions. In chick embryo, endodermal ECM has a precise spatial and temporal organization to stimulate and guide mesodermal cell migration [27]. In particular, the fibronectin, abundant between the precardiac mesoderm and the adjacent endoderm [28], is expressed in a localized anterior patch to instruct correct cell migration. Not all cells respond to this stimulus, but only anterior cells are required to form strong adhesive regions with fibronectin. Thus, the contractile cytoskeletal activity of the responsive cells can generate forces propagated throughout the layer of connected cells to pull them in the proper direction [27]. As an alternative, in keeping with the findings by Newgreen and Thierry in cranial neural crest cell migration, deposition of fibronectin by the migrating ‘leader’ cells may form fibronectin gradients to guide trailing cells in their migratory paths [29], thus making the process pulling forces-independent.

The mechanical interaction between cells and ECM is important for proper cell allocation, but also for cardiac differentiation during heart morphogenesis. For example, a progressive increase in matrix stiffness in the heart primordium has been related to the initial beating of the embryonic myocytes through the effect of matrix stiffness on the coordinated opening of the mechanosensitive Ca2^+^ channel before the onset of electromechanical coupling [30]. This suggests that, in the early embryonic heart tube, the first signaling pathway for induction of the coordinated myocyte beating has a pure mechanical rather than electromechanical origin, strictly related to stiffening of the cardiac matrix [31]. The mechanical changes occurring in consequence of cardiac matrix maturation may also have important readouts for the acquisition of the adult cyto-architecture of the myocytes. For example, the hardening of the cardiac matrix occurring after birth is involved in the drastic change in the phenotype of myocytes, with definitive cell maturation, modification in their cyto-architecture and shape, from polygonal to rod-like, and preferential alignment in bundles [32,33]. These signals may be finally linked to the sudden mitotic arrest occurring after birth, due to hindrance of the sarcomere structures for the formation of the mitotic spindle.

Other forces, such as shear stress and cyclic strain are finally important for cardiac progenitor differentiation in vitro. For example, mouse embryonic stem cells differentiate in ectodermal or mesodermal lineage depending on the magnitude and the time exposure to fluid shear stress [34]. Moreover, cyclic strain enhances the cardiac specific gene expression in embryonic stem cells and induces embryonic stem cells-derived cardiomyocytes hypertrophy with improvement of their contractile function [35,36].

Taken together, these evidences establish the importance of mechanosensation for the overall developmental process of the heart, from primary differentiation of the most primitive cardiac mesoderm progenitors, to the end of the myocyte hypertrophic growth, occurring after birth.

### 2.2. Role of Mechanical Factors in Myofibroblast Activation and Cardiac Fibrosis

Heart failure is a clinical condition subject to continuous growth due to the elevation of life expectancy worldwide. Myocardial maladaptive remodeling is one of the earliest hallmarks of heart failure, a pathology characterized by inflammation and a progressive fibrosis that negatively affects the contractility of the myocardium, and leads to replacement of myocytes with a stiff fibrotic tissue [37]. In healthy conditions, quiescent cardiac fibroblasts are responsible for renewal of ECM proteins [38]. Under pathological/risk conditions (ischemia, mechanical overload, aging), these cells become activated, proliferate and differentiate into myofibroblasts, contractile and matrix remodeling cells [39]. These cells respond to inflammatory signaling, and induce matrix degradation and collagen deposition, leading to enhanced mechanical loading and stress. This process, in concert with ECM crosslinking, imbalanced composition and remodeling [40], determines a significant increase in myocardial stiffness (from 10–20 kPa Young’s Modulus to 50–200 kPa), which cooperates with profibrotic stimuli (e.g., TGF-β) to determine a chronic scarring process [41].

Cardiac fibroblast activation is affected by mechanical cues through direct translation of cytoskeleton tensioning into intracellular activation cascades or by controlling the release of paracrine signals [42]. For example, the persistent elevated stretch of mice cardiac fibroblasts, mimicking the mechanics of the infarct region, stimulates sustained production of ECM, causing myocardial stiffening [43] (Figure 3). Moreover, matrix hardening associated with infarct scar maturation induces cell spreading, formation of smooth muscle α-actin stress fiber and expression of collagens I/III [42,43]. The complex YAP/TAZ, involved in cell mechanosensing [44], exerts a crucial role in adult cardiac fibroblasts migration, proliferation and differentiation [45]. For example, in adult murine heart, cells with nuclear YAP/TAZ localize at the border zone of the ischemic areas, suggesting a prompt response of resident stromal cells to ischemia in the post infarcted myocardium and a role in collagen deposition and stiffening of myocardial matrix [45]. These findings suggest the existence of a mechanical control of cardiac fibroblast behavior that could be involved in the pathological evolution of cardiac fibrosis disease (Figure 3).

The mechanosensing-dependent pro-fibrotic activity of cardiac fibroblasts in the failing heart may be hampered by metabolic insults, epigenetic changes and inflammatory stimuli. This consideration emerges from experiments showing that mesenchymal stem cells exhibit “mechanical memory” effects [46]. In particular, cells cultured in contact with high stiffness substrates appear the gene expression setting acquired in the stiff environment, even after shifting into a soft environment. Thus, similar to the so-called epigenetic memory associated with the exposure of cells to altered metabolic conditions (e.g., hyperglycemia) [47], tissue mechanics could also irreversibly affect the expression of pathology-related genes, even after reversion of matrix mechanics to normal conditions. In keeping with this hypothesis, there is the finding that migratory epithelial cells primed on hard ECM for a defined duration migrate faster and display higher actomyosin expression compared to controls even when in contact with a softer matrix [48]. A similar behavior was observed in lung myofibroblasts cultured on substrates with elastic modulus corresponding to pathologically stiff fibrotic tissue [49].

Findings obtained in tumor biology show that there is an interplay between cell metabolism and tumor microenvironment [50,51]. For example, alterations of metabolism can affect the mechanical property of tumor ECM by modifying Proline metabolism and collagen synthesis [52], thus contributing to ECM stiffening in various human solid tumors [53]. Analogously, metabolic dysfunctions related to fibrotic response in different cardiac pathology models [54,55] may alter the mechanics of the extracellular matrix [56] contributing to establish pro-fibrotic gene expression signatures in activated myofibroblasts. The mechano-dependent alterations in gene transcription may thus translate into permanent epigenetic changes underlying at first transient, and thereafter definitive alterations in cell functions contributing to chronic fibrosis [19,57]. In keeping with this hypothesis, mechanical cues and matrix stiffness can regulate chromatin remodeling and the global epigenetic state affecting DNA methylation and histone modifications [58]. These alterations may be finally sustained by classical pro-fibrotic signaling. For example, TGFβ, a potent mediator of myofibroblast activation [59], induces deposition of H3 lysine 4 (H3K4) marks on profibrotic genes in liver fibroblasts [60].

Altogether, these evidences suggest the existence of ‘mechano-paracrine’ gene expression circuitries establishing altered behavior of cardiac fibroblasts during fibrosis setting. These conditions may permanently alter the epigenetic landscape of the myofibroblasts leading to irreversible myocardial scarring. Future studies will be necessary to substantiate these hypotheses mechanistically and to assess whether interfering with permanent activation of mechano-paracrine signaling is a viable way to reduce the impact of cardiac fibrosis.

## 3. Cell-Based Mechanosensing Controls Valve Formation and Disease

### 3.1. Hemodynamics Drive Cardiac Valve Morphogenesis

Heart valves are very specialized structures in the body. They ensure an efficient, unidirectional flow of the blood through the heart cavities. Similar to the development of myocardial tissue, cell mechanics play a role in the development of heart valves through extensive remodeling processes of the valve primordia and patterning/formation of functional leaflets. Perturbations in these mechanisms lead to valve malformations and congenital heart defects.

Different forces generated from the developing heart are involved in valve morphogenesis. Before valve development, there is a considerable blood regurgitation resulting from reflow of the blood from the ventricle to the atrium [61]. In zebrafish, this flow pattern might be responsible for the beginning of valve morphogenesis by instructing endocardial cells in the atrioventricular canal (AVC) towards valvulogenesis [62]. Indeed, the first step in valve development is the clustering of endocardial cells to form an endothelial ring lining AVC [63]; in this morphogenetic event, blood flow is not necessary for ring formation, but it is crucial for inducing cell shape changes and leaflet invagination [62]. Different reports demonstrated that reversing flows initiate the invagination process by inducing cytoskeletal rearrangements of the endocardial cells in the AVC to build a functional valve [62,64]. Among different transcription factors involved in the blood flow-dependent morphogenetic process, *klf2a*, a known atheroprotective gene [65], was found to regulate cell morphology and proliferation in the endocardium through the activation of Trpv4, a mechanosensitive ion channel specifically expressed during heart valve development [66]. Zebrafish Trpv4 mutant embryos present dramatic valve defects, due to an abnormal cellular re-organization during valve formation, causing impaired valve leaflet development [66].

In mammalians, the initial step of the valve formation process is different from zebrafish as endocardial cells undergo an endocardial-mesenchymal transition (Endo-MT) to invade the ECM and form the cardiac cushion tissue [67]. Mutations in genes encoding for factors involved in ECM composition or cellular signaling molecules severely impair this process [68,69,70], preventing cells from reaching their mature phenotype and forming valve leaflets and chamber septation. As for the differentiation of primary cardiac progenitors, mechanical forces such as shear stress can contribute to Endo-MT of endocardial cells to form valve leaflets [71].

### 3.2. Mechanoperception and Calcific Aortic Valve Disease

Calcific aortic valve disease (CAVD) is a pathological condition that starts from mild valve thickening without obstruction of blood flow (aortic sclerosis), and ends with a severe calcification and impaired leaflets functionality (aortic stenosis) that require valve replacement [72]. The cells involved in this maladaptive remodeling are the so-called “valve interstitial cells” (VICs), a heterogeneous population of quiescent fibroblasts involved in ECM renewal [73]. CAVD is associated with pathological differentiation of VICs into activated myofibroblasts. These cells remodel the valve tissue by an increased deposition of ECM proteins, such as collagen, elastin and glycosaminoglycans, and overexpression of matrix metalloproteases [74]. Beyond the inflammatory component, mechanical stimuli likely contribute to initiation and progression of this disease (Figure 3). For example, it was noted that there was a strong correlation between the leaflet areas where the calcific lesions form preferentially and the regions of the leaflets (typically the ‘belly’ portions and the commissures) where mechanical stress reaches maximal values [75]. Moreover, calcific lesions of the aortic valve preferentially develop in regions exposed to disturbed blood flow [76] or subjected to high shear and bending stress [77,78]. They further affect predominantly the fibrosa layer, which is the stiffest layer of the leaflet due to its prevalent collagen composition [79,80]. The mechanosensitive characteristics of VICs have been demonstrated with the findings that these cells are able to “sense” the local mechanical proprieties of the ECM, such as the stiffness [81]. These responses are mediated by the tensioning of the cytoskeleton, and this is associated with the increase in rigidity of the cells themselves, as assessed by nano-indentation (e.g., atomic force microscopy) methods [82] (Figure 1).

Mechanistically, the importance of substrate compliance in mesenchymal cell differentiation has been generally demonstrated by establishing a link between the cytoskeleton tensioning and the activity of the YAP/TAZ transcriptional coactivators [83]. The implication of YAP/TAZ mechano-dependent signaling in pathological evolution of the aortic valve has been characterized in a study from our group [84]. This research showed that human VICs, obtained from stenotic and insufficient valves and adhering onto substrates with controlled stiffness, exhibited stiffness-related YAP nuclear localization, strictly dependent on RhoA/ROCK-mediated cytoskeletal tensioning. In particular, stenotic VICs showed increased YAP-dependent transcriptional activity in conjunction with increased cytoskeletal tensioning, and this could be reverted by using an inhibitor of the ROCK-dependent pathway (Y27632) [85]. In summary, matrix elasticity induced a rearrangement in the cytoskeleton, which resulted into specific changes in gene expression through the activity of YAP/TAZ complex, with consequences for cell differentiation and pro-fibrotic commitment. This suggests that the hardening of the ECM occurring in response to leaflet tissue inflammation activates VICs via a mechanical-dependent pathway and this promotes their evolution into the calcific phenotype (Figure 3). In keeping with this hypothesis, VICs mineralization involves RhoA/ROCK pathway [86] with active deposition of calcific nodules, a hallmark of valve leaflets calcification [48,87].

## 4. Mechanical Stimuli Impact on Blood Vessels Morphogenesis and Pathology

### 4.1. Mechanical Forces Prime Early Vascular Development

As we discussed in another contribution [88], hemodynamic forces have an important role in the development of primitive vessels. For example, after early specification of the endothelial lineage in the yolk sac vasculature, primitive vessels undergo remodeling in the presence of the flow forming a branched, hierarchically organized network of large and small-caliber vessels that deliver blood to the embryo [89]. Paradoxically, at the earliest developmental stages, the mechanical component of the blood flow could be even more important than the oxygen and nutrient supply for the tissues. In fact, several studies show that blood flow is absolutely required for vessel and cardiac morphogenesis independent of trophic function [90,91,92]. For example, in zebrafish exposure to carbon monoxide of the embryos did not affect the initial stages of vasculogenesis [93], suggesting that the function of the blood flow is not necessary for delivering oxygen during the shaping of primitive vascular systems. It has been hypothesized that blood circulation could activate cell-signaling cascades in endothelial cells (ECs), forming the primitive yolk sac vascular plexus [94], thus triggering the proper vascular development. Supporting this hypothesis, the findings of Lucitti and coworkers [95] demonstrated the role of fluid-derived force in vascular remodeling in mice. In particular, they observed that blood flow with low hematocrit, and therefore a reduced viscosity, is not sufficient to induce vascular growth, arguing against the idea that this process is triggered by soluble factors distributed by blood flow. Increasing the viscosity was in fact enough to rescue the remodeling deficiency in low-hematocrit embryos, suggesting an important role for erythroblasts in increasing the impact of circulatory flow in the yolk sac. Shear stress, which is dependent on the blood viscosity, might be the critical force acting on ECs of vascular plexus, activating mechano-signaling pathways and driving the cell differentiation. One of the signaling pathways involved in mechanical activation of vessel morphogenesis is that mediated by endothelial nitric oxide synthase (eNOS). Shear stress acts in fact on ECs mechanoreceptors, activating the production of nitric oxide (NO) by eNOS, and regulating vascular tone [96,97].

Arterial and venous endothelial cells are characterized by different gene expression signatures even before the onset of blood flow. This suggests that arterial and venous specification is genetically ‘hard-wired’ [98]. On the other hand, blood flow has a crucial role in maintaining separation between these identities [99]. Fluid shear stress induces arterial differentiation of endothelial progenitor cells, by upregulating the mRNA levels of arterial markers (e.g., ephrinB2, Notch 1/3) and decreasing the expression of venous endothelial cell markers (e.g., EphB4). Another evidence shows that shear stress activates intracellular signaling pathways in endothelial progenitors, including p38 and ERK1/2, increasing cell differentiation toward arterial phenotype [100]. During vessel development and maturation, vascular smooth muscle cells (VSMCs) are also involved in different processes as biosynthetic, proliferative and contractile components of the vessel wall. Indeed, as blood flow begins, VSMCs are recruited from the surrounding mesenchyme and cardiac neural crest to produce and organize ECM proteins [101]. They proliferate after the association with the nascent endothelium [102] and, at the end of vasculature maturation, they control the vascular tone, peripheral resistance and distribution of blood flow in the whole body. Differentiation of VSMCs occurs concomitantly with the synthesis of specific ECM proteins necessary to form circumferential layers around the endothelial tubes. In particular, the expression of fibronectin is crucial for the early migratory events of vessel wall development, while the content of laminin is correlated to the extent of vessel wall maturation [103].

Vascular ECM is not an inert supporting network. In fact, vascular cells are able to detect changes in matrix rigidity and composition during tissue remodeling with consequence for gene expression proliferation and differentiation [104]. For example, stiffness of the matrix affects lineage differentiation of embryonic stem cell-derived vascular progenitors (VPCs) with a preferential endothelial commitment on softer hydrogels, and SMCs phenotype onto high stiffness substrates [105]. In addition, ECM development and VSMCs differentiation are correlated with an increase in pulse pressure in the embryonic circulation, and differentiation of presumptive VSMCs surrounding the endothelium [106]. One of the signaling pathways involved in this mechano-dependent differentiation process is the Notch/Hippo pathway. In particular, mutations in components of the Notch signaling causes defects in the cardiac outflow tract [107]. For proper functioning of this regulator, Hippo pathway effectors, YAP and TAZ, are also required. Indeed, neural crest-specific deletion of YAP and TAZ, gives rise to precursors that are able to migrate, but fail in VSMCs differentiation [108]. Taken together, these evidences suggest that changes in the embryonic blood flow contribute to the generation of signaling cascades involved in vessel wall morphogenesis through transmission of mechanical stimuli.

### 4.2. Mechanical Strain Induces the Activation of Pro-Pathologic Pathways in Large Vessels

During adult life, arteries undergo a continuous remodeling of their matrix to maintain optimal elasticity and tone, and keep a physiologic patency. It has been reported that endothelial cells are highly responsive to modifications in flow patterns, and this maintains the fluid shear stress within a desired range [109]. In case of high or low shear stress, pathways regulating vessel remodeling to restore the original level of shear stress become activated [109]. Shear stress is also able to define the endothelial phenotype. Indeed, the zinc finger transcription factor lung Krϋppel-like factor-2 (*klf2*), expressed in adult vasculature endothelial cells, is involved in this mechano-dependent gene transcription regulation by inducing endothelial phenotype in response to flow [110]. The increased expression of *klf2* in high-shear stress regions in murine carotid artery determines an up-regulation of eNOS and a decrease in endothelin-1 and adrenomedullin expression, confirming its role in vascular tone control, in particular in vessel vasodilation [111]. Variations in shear stress are sensed by mechano-receptors (e.g., integrins), and downstream signaling cascades are activated to induce rapid changes in cytoskeleton structure and triggering specific gene-expression programs [112,113]. The consequence is the synthesis/release of vasoactive mediators (e.g., NO, prostaglandin) reducing shear stress, of ECM remodeling enzymes (e.g., lysyl oxidases) promoting vascular wall repair, and of growth factors (e.g., PDGF, bFGF) controlling SMCs survival and proliferation [114,115]. This is a compensatory process needed to maintain an anti-proliferative, anti-thrombotic and anti-inflammatory phenotype of the endothelium.

Atherosclerotic plaque formation occurs in arteries at specific sites where perturbed flow or a low shear stress predominates (e.g., the concave regions of arterial bends; branching points). These conditions favor vascular inflammation, increased endothelial permeability, ROS generation and expression of receptors or cytokines promoting homing of leukocytes [116]. The susceptibility to atherosclerosis correlates with failure of endothelial cells to align in direction of the flow [117]. In fact, when ECs are not aligned along the major flow direction inflammatory pathways become activated [117]. In particular, this involves activation of the NF-кB transcription pathway, increased ROS production, and reduced NO production due to eNOS downregulation.

A hypothesis for intracellular transduction of flow-related mechanical stimulation involves the function of the primary cilia. Primary cilia are cellular protrusions composed of microtubules connected to cytoskeleton. Endothelial cells have abundant cilia in regions subjected to low shear stress or disturbed blood flow, while they are absent in regions with high laminar shear stress [118]. In ApoE-deficient mice, primary cilia are located upstream and downstream of atherosclerotic lesions, matching the locations of plaque rupture and inflammation. This suggests that atherosclerosis may be associated with cilia mechanosensing activity [119]. Variations in shear stress can be sensed by cilia through mechanosensitive calcium channels, such as polycystins 1 and 2 [120]. Since the basal body of cilium is physically connected to the cytoskeletal microtubules, movement of the cilium on endothelial cells induces cytoskeletal rearrangements responsible for the initiation of signaling cascades characterized by an increased intracellular Ca^2+^, production of NO and vasodilation responses [121]. Another interesting correlation between atherosclerosis and disturbed flow patterns- could be at the level of the ECM. Injury or inflammation cause the degradation of the EC basal membrane and deposition of transitional ECM proteins such as fibronectin and fibrinogen [122]. In particular, these matrix proteins are deposited in regions of disturbed flow in atherosclerosis-prone arterial regions, where they promote the activation of NF-кB inflammatory pathway leading to atherogenesis [123,124]. Even once the atherosclerotic plaque is formed, mechanical forces may contribute to maintain the pathologic condition. For example, it has been hypothesized that forming plaques may create sections of the vessels with a disturbed flow and this may result in plaque rupture [125,126]. Furthermore, increased expression of metalloproteinases, due to disturbed flow patterns, determines a collagen degradation of the fibrous cap incrementing the plaque vulnerability. Vessel wall remodeling also occurs as adaptation to an elevated pulsatile flow, which promote ECM-regulated SMC migration and proliferation. A prerequisite for SMC migration is the degradation of ECM through secretion of metalloproteases (MMPs) from SMC [127]. Hemodynamics variations are known to regulate MMPs expression and activation [128]. Blood flow may have finally an important function in MMP-dependent remodeling. Indeed, in a murine model of blood flow cessation, interruption of the flow in carotid arteries caused MMP-9 upregulation and arterial enlargements, and this was reverted using a nonselective MMP inhibitor [129,130].

Not only shear stress, but also increased transmural pressure due to hypertension activates MMP-mediated degradation of ECM in the medial layer, leading to tissue hypertrophy [131]. This suggests that oscillatory pulsatile pressure also plays a crucial role in SMCs function. In particular, SMCs sense this oscillatory pressure by cyclic compression and strain forces, depending on their geometrical position inside the vessel. The orientation of SMCs is not fixed but adaptable in response to the magnitude of strain [132]. For example, this happens in saphenous vein bypass grafts, where vein conduits shift from a low and constant flow in venous environment to a high and pulsatile artery circulation [133]. Under ‘arterialization’ conditions, veins are subjected not only to high flow shear stress, but also to cyclic wall strain due to coronary flow pattern. This has consequences both for wall structure and for the SMC phenotype [134]. Cyclic strain, as well as biochemical factors, induces a phenotype switching in SMCs, from the contractile state to the synthetic/proliferative (Figure 3). This process is a hallmark of intimal hyperplasia in saphenous vein grafts [135]. Variations in the composition of ECM also change the response of SMCs to mechanical strain, suggesting that specific patterns of matrix-integrin engagement orchestrate downstream intracellular signaling pathways [136]. Also in this case, the mechanosensors YAP and TAZ seem to be involved by inducing SMCs migration and proliferation. YAP inhibits the activity of the promoter of Hic-5 and smooth myosin heavy chain, and induces the expression of TEAD target genes, such as cyclin D1, controlling cell proliferation. Rho inhibition or actin cytoskeletal disruption prevents YAP/TAZ activation in vascular SMCs [137]. In addition, YAP suppresses the expression of SMC contractile genes by interacting with the myocardin-SRF complex [138].

SMCs are not the only players in vascular diseases. In fact, adventitial fibroblast cells (AFs) may play an important role for neointima formation [139]. For example, in porcine models of balloon injury, AFs are converted into myofibroblasts and are involved in the wall repair process through abundant ECM deposition, responsible for adventitial thickening [140]. AFs phenotypic switching occurs at 2 days after the injury, and this is followed by increased migration from the adventitia to the medial layers where they acquire a ‘myo-fibroblast’ phenotype [141,142] (Figure 3). Adventitial migration is independent of the proliferation rate, and it is activated by specific MMPs or inhibited by their tissue inhibitors (TIMPs). Specific factors such as the TGF-β [143] are also mechanically induced, acting as a chemoattractants for these cells. As an example, in a study from our laboratory, we found an upregulation of the TGF-β signaling pathway in human saphenous veins stimulated ex vivo with a bioreactor system mimicking arterial-like pressure [128].

Another pro-pathological factor correlating to altered mechanical forces is aging. Stiffening of the vascular network is considered as hallmark of normal aging and is associated with systolic hypertension, coronary artery disease and atrial fibrillation [144]. In older individuals, blood vessels stiffen through modifications in ECM composition and calcification, reducing elasticity of the vessels and causing hypertension. Among the mechanisms responsible for arterial stiffening, there is an increased mechanical stress due to elevated pressure that leads to structural disorganization and fragmentation of elastic fibers [145]. Arterial media calcification and increase in arterial stiffness may have an effect on pathologic activation of endothelial cells through endothelial-mesenchymal transition. In this process, matrix stiffness seems correlated to epigenetic regulation of SMCs phenotype by downregulating DNA methyltransferase [146]. This results in a hypercontractile SMCs population with osteochondrogenic features able to produce calcifying exosomes. At molecular level, matrix stiffness activates integrin-FAK pathway thus favoring the activation and nuclear translocation of YAP/TAZ complex and transcription of genes involved in the calcific process (e.g., RUNX2, BMP2) (Figure 1). In addition, a decreased NO production activates the RhoA/ROCK pathway, resulting in nuclear translocation of YAP/TAZ, interaction with EndMT inducers Snail, Slug and Twist and induction of Runx2-dependent fibrosis and ECM mineralization [147].

## 5. Conclusions

Mechanical forces do not only play a central role in cardiac morphogenesis and tissue patterning, but are also important for cardiovascular disease progression. Although this conclusion was anticipated in pioneering investigations, systematic dissection of molecular pathways activated by mechanical stresses has only recently been endeavored. Cells are able to “sense” the normal/altered mechanical environment and, through cytoskeleton rearrangements, transfer these signals into nucleus for gene expression regulation. How the highly conserved mechanical stimuli are able to drive the embryo development, and affect tissue remodeling in cardiovascular pathologies is not entirely characterized. However, a better understanding of these mechano-dependent mechanisms will increase the chances of reversing developmental defects, and establish new ways for treating cardiovascular fibrotic disorders based on targeting of mechano-dependent intracellular pathways.

## Figures and Tables

**Figure 1 cells-08-01607-f001:**
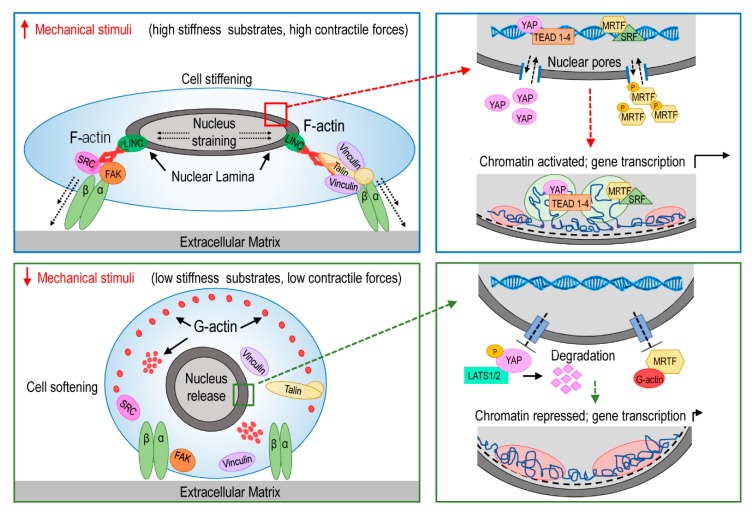
Effect of mechanical forces on activation of transcriptional circuitries. Top panels: in the presence of contractile/stretching forces, or when in contact with hard adhesion substrates, cells (e.g., fibroblasts) transduce mechanical signaling through focal adhesion contacts and F-Actin cytoskeleton. Traction forces are transmitted through proteins of the nuclear lamina to the nucleus with consequent opening of nuclear pores and nuclear translocation of mechanosensing-dependent transcription factors (e.g., YAP/TAZ and MRTF transcription factors). Chromatin activation can then occur with consequent increase in target genes expression. Bottom panels: when cells are not mechanically stimulated, or are in contact with soft matrices, assembly of focal contacts is less efficient and proteins involved in the polymerization of stress fiber do not transduce mechanical deformation to the nucleus. Under these conditions, the nuclear pores are closed and this prevents chromatin activation and reduces the expression of mechanosensing-dependent genes.

**Figure 2 cells-08-01607-f002:**
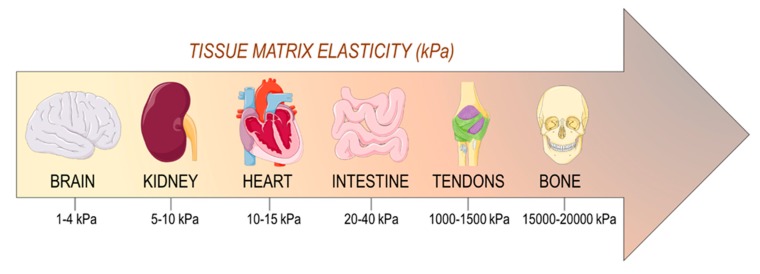
Stiffness range (Young’s modulus, kPa) as assessed in a variety of human tissues. Data derived from [11,12].

**Figure 3 cells-08-01607-f003:**
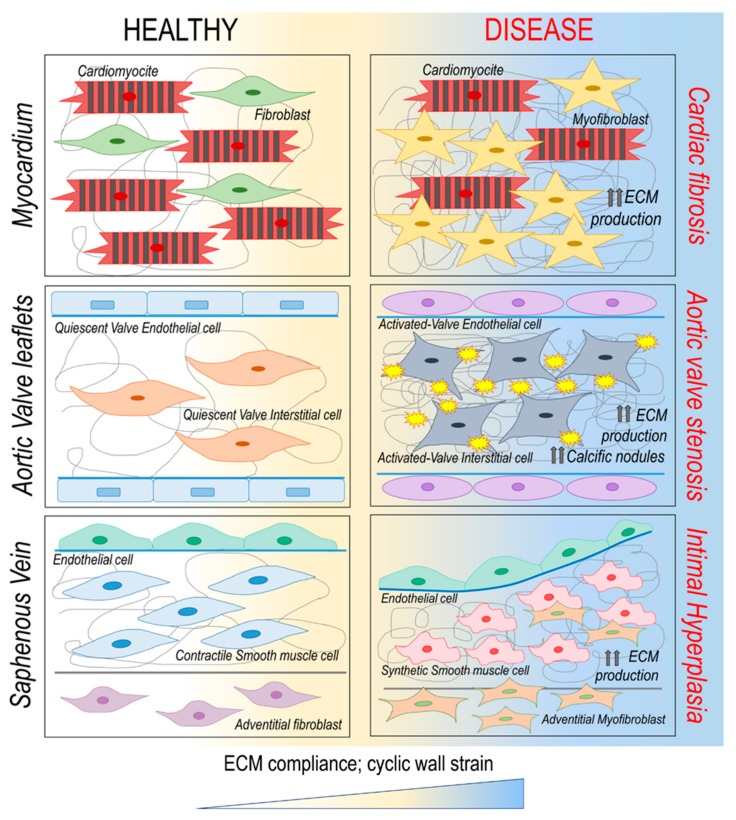
Implication of matrix remodeling on change in the mechanical characteristics of the extracellular matrix, and the relative cellular strain, in three major cardiovascular pathologies (cardiac fibrosis, aortic valve stenosis and intima hyperplasia in vein graft disease). A common trait is represented by the activation of quiescent stromal cells and their differentiation into myofibroblasts.

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
