# Peer review of "Mechanotransduction in the Cardiovascular System: From Developmental Origins to Homeostasis and Pathology"

_cells, 2019, doi:10.3390/cells8121607_

Round 1

Reviewer 1 Report

This manuscript provide  an overview of the influence of mechanotransduction in the circulatory system both in embryos and in disease. Although potentially helpful, several additions and reparations/rephrases are needed. These are highlighted below.

The definition of substrate characteristics is rather binary in terms of hard and soft (see Fig 1) and therefore needs further explanation as there exist many substrates from bone to hyaluronic acid and many in between. Please give examples of tissues with various characteristics and gradations of stiffening to softening, as in the right panels of Fig1 an all-or-none machinery is depicted. Several genes may be activated/inhibited in intermediate circumstances, accounting for various circumstances during development and disease progression. Line 110/1111 “coherent epithelia”, to my knowledge the cells in the cardiac crescent do not conform to the definition of epithelium (e.g. apical-basal polarity, basement membrane, tight junctions and more) and therefore could just be called cardiac crescent. Line 112/117 It is unclear from the description whether endoderm or mesoderm is the migratory component, please rephrase. Line 120-123. If I remember correctly an alternative hypothesis has been forwarded by Newgreen and Thierry (1980) during their study of cranial neural crest cells, a similar model of early migratory cells. The leading NCC produce fibronectin, the trailing cells (not producing FN) following the leaders. This makes pulling forces unlikely. I suggest that this will be discussed. Line 125 Lineage separation between germ lines takes place well before cardiac function. The latter is carried out by the cardiac tube that is mostly separated from the environment (except the endocardium) as it is located and freely moving in the pericardial cavity. In chicken embryos this is typically ± HH9. Of course, the tube connects to the venous and arterial pole. Therefore, cyclic shear stress and strain can only be conveyed through the poles, but only much later in development compared to germ line separation. Line 133 Please provide a hypothesis how early myocytes can react to a stiffening matrix by contracting while, “other embryonic cells” do not show this behavior. Line 139/140 “after birth” is far beyond the initial processes described shortly before, also here: a wrong timing as in #5. Furthermore describe “hardening” as related to Fig1 Infarction is primarily induced by lack of oxygen, followed by a number of hypoxia-related processes, these should also be discussed. Line 207 “most-specialized” is an overstatement. Line 214. In contrast, many papers describe minimal regurgitation due to the presence of endocardial cushions, the predecessors of the valves, see for instance Ursem NTC et al. (2001). Line 235 and further, for the AV valves and pathology I advise to study and incorporate papers by e.g. Peterson and Deruiter (2018) for the cellular and genetic composition of the (bicuspid) Ao valve, Mahler and Butcher (2014) for the hemodynamics, and seminal work of M.S Sacks on the stiffness of the (adult) Ao valve. Line 272 Very early vasculogenesis is defined by the interaction of endothelial stalk and tip cells as extensively studied in the developing retina and zebrafish embryos. It is hard to conceive that in these blind-ending sprouts hemodynamic factors are involved. Please take this into consideration in discussing this aspect.

Author Response

Point-by-point response to REVIEWER 1:

“This manuscript provide an overview of the influence of mechanotransduction in the circulatory system both in embryos and in disease. Although potentially helpful, several additions and reparations/rephrases are needed”

We thank the Reviewer for this general comment. In our revised manuscript version, we have thoroughly revised the text and addressed the specific concerns, where appropriate. With these changes, we hope to have improved the overall impact of our manuscript.

The definition of substrate characteristics is rather binary in terms of hard and soft (see Fig 1) and therefore needs further explanation as there exist many substrates from bone to hyaluronic acid and many in between. Please give examples of tissues with various characteristics and gradations of stiffening to softening, as in the right panels of Fig1 an all-or-none machinery is depicted. Several genes may be activated/inhibited in intermediate circumstances, accounting for various circumstances during development and disease progression.

We agree with Reviewer that compliance of the tissues is never a binary value (stiffness gradients actually exist), and that the definition of ‘soft’ and ‘hard’ can be used just in relative and not absolute descriptions of tissue mechanics. Since as correctly pointed out by Reviewer, there are different reference compliance values in different tissues in the human body, we have introduced a new figure (new Figure 2), where we report the reference stiffness values of various organs/tissues, as described in the literature. We also added text (lines 67 -76) as a description of the new figure.

“Line 110/1111 “coherent epithelia”, to my knowledge the cells in the cardiac crescent do not conform to the definition of epithelium (e.g. apical-basal polarity, basement membrane, tight junctions and more) and therefore could just be called cardiac crescent”

The necessary amendment of the text to adhere to correct definition has been done at line 122, thank you.

“Line 112/117 It is unclear from the description whether endoderm or mesoderm is the migratory component, please rephrase”

We have rephrased the text where necessary (line 125 and line 129), thank you.

“Line 120-123. If I remember correctly an alternative hypothesis has been forwarded by Newgreen and Thierry (1980) during their study of cranial neural crest cells, a similar model of early migratory cells. The leading NCC produce fibronectin, the trailing cells (not producing FN) following the leaders. This makes pulling forces unlikely. I suggest that this will be discussed.”

We have added the suggested reference and discussed the alternative model for the early cardiac mesoderm migration, thank you (lines 134 – 137).

“Line 125 Lineage separation between germ lines takes place well before cardiac function. The latter is carried out by the cardiac tube that is mostly separated from the environment (except the endocardium) as it is located and freely moving in the pericardial cavity. In chicken embryos this is typically ± HH9. Of course, the tube connects to the venous and arterial pole. Therefore, cyclic shear stress and strain can only be conveyed through the poles, but only much later in development compared to germ line separation”

We thank Reviewer for this point, which probably arises from an incorrect wording of our original sentences. We have now rephrased to explain that mechanical interactions with the matrix are not the only ones that are involved in migration of cardiac mesoderm cells, but also for primary differentiation of myocytes. And that other forces such as shear stress and cyclic strain, are also involved in differentiation of primitive cardiac progenitors, e.g. in vitro (lines 138 – 160).

“Line 133 Please provide a hypothesis how early myocytes can react to a stiffening matrix by contracting while, “other embryonic cells” do not show this behavior”

This hypothesis has been made in references 30 – 31, where the increased stiffness of the extracellular matrix surrounding the early myocyte progenitors has been correlated with an increased coordination of mechano-perceptive ion channels opening by the increasing stiffness. This hypothesis has been better clarified in the revised manuscript (lines 139 – 145).

“Line 139/140 “after birth” is far beyond the initial processes described shortly before, also here: a wrong timing as in #5. Furthermore describe “hardening” as related to Fig1 Infarction is primarily induced by lack of oxygen, followed by a number of hypoxia-related processes, these should also be discussed”

We introduced the concept of progressive hardening of the cardiac matrix as a continuous process that drive fundamental changes in the myocardium characteristics to express the concept that mechanosensation is important for the overall evolution of the cardiac tissue, from its initial specification to its final maturation. We are certain that Reviewer will appreciate our effort at establishing this conceptual continuum. In our manuscript, we did not intend to describe hardening as a direct effect of hypoxia, but an indirect consequence that originates from the lack of oxygen or other concurrent risk conditions. This is clearly stated in the revised manuscript lines 167 – 173.

“Line 207 “most-specialized” is an overstatement”

We have corrected, thank you (line 220).

“Line 214. In contrast, many papers describe minimal regurgitation due to the presence of endocardial cushions, the predecessors of the valves, see for instance Ursem NTC et al. (2001). Line 235 and further, for the AV valves and pathology I advise to study and incorporate papers by e.g. Peterson and Deruiter (2018) for the cellular and genetic composition of the (bicuspid) Ao valve, Mahler and Butcher (2014) for the hemodynamics, and seminal work of M.S Sacks on the stiffness of the (adult) Ao valve”

To address Reviewer’s concern and avoid confusion, we have avoided referring to flow perturbations in the presence of endocardial cushions (line 223, revised manuscript), thank you. Concerning the second part of the concern, we introduced the references from Peterson and Deruiter (2018) and from Mahler and Butcher (2014). These introductions have been made in the revised manuscript at lines 240 – 246.

Line 272 Very early vasculogenesis is defined by the interaction of endothelial stalk and tip cells as extensively studied in the developing retina and zebrafish embryos. It is hard to conceive that in these blind-ending sprouts hemodynamic factors are involved. Please take this into consideration in discussing this aspect.

We have rephrased our sentence where text might have been misleading, thank you (lines 293 – 294).

Reviewer 2 Report

This is a very thorough review on the role of mechanotransduction and mechanosensing for cardiac development and homeostasis.  Figure 1 is very useful.  Additional Figures to augment key points would increase the value of this Review article.   Overall, I do not have suggestions for additional content and primary article review.  I do however recommend editing for proper English and typographical errors.  With that, I recommend the following changes:

Ln 21:  Replace permanent or with long-lasting (permanent seems too strong)

Ln 23:  Replace permanent with persisting (again permanent seems too strong)

Ln54-61:  Significant overuse of e.g.  Its distracting to the reader.  This needed editing.

Ln77:  replace ‘they’ with the more direct ‘cell forces’

Ln107:  replace ‘four fibrous valves’ with ‘associated fibrous valves’  (not all experimental animal models have 4 valves).

Ln112:  Awkward wording:  ‘It has been demonstrated an active mechanical role for the endoderm during this process’:  This sentence needs editing.

Ln122-23:  Replace ‘and pulls’ with ‘to pull’

Ln130:  Replace ‘The particular’ with ‘Specific’  (particular is redundant with the next sentence)

Ln137:  Delete (‘also’)

Ln166:  Delete ‘localized’ (redundant with localize)

Ln177:  Change: ‘also tissue mechanics could irreversibly’ to ‘tissue mechanics could also irreversibly’

Ln209:  Change: ‘cell mechanics plays’ to ‘cell mechanics play’

Ln 230:  Replace: ‘cells undergo to endocardial-mesenchymal transition’ with ‘cells undergo an endocardial-mesenchymal transition’

Ln258:  Replace: ‘been in particular observed in’ with ‘been characterized by’

Ln259:  Break up the sentence: ‘ …a study from our group.  This research showed that human VICs, …’

Ln289:  Delete ‘from’:  use ‘rescue the remodeling’

Author Response

Point-by-point response to REVIEWER 2:

“This is a very thorough review on the role of mechanotransduction and mechanosensing for cardiac development and homeostasis.  Figure 1 is very useful.  Additional Figures to augment key points would increase the value of this Review article”

We thank the Reviewer for this general comment. In our revised manuscript version, we have introduced two new figures to enhance the impact of our manuscript.

“Overall, I do not have suggestions for additional content and primary article review.  I do however recommend editing for proper English and typographical errors.  With that, I recommend the following changes”

We have added modifications in the revised manuscript as suggested. Below the specifications of the text lines corrected in the revised manuscript.

Ln 21:  Replace permanent or with long-lasting (permanent seems too strong)

Corrected in line 21. Thank you.

Ln 23:  Replace permanent with persisting (again permanent seems too strong)

Corrected in line 23. Thank you.

Ln54-61:  Significant overuse of e.g.  Its distracting to the reader.  This needed editing.

Corrected in lines 56 - 61. Thank you.

Ln77:  replace ‘they’ with the more direct ‘cell forces’

Corrected in line 87. Thank you.

Ln107:  replace ‘four fibrous valves’ with ‘associated fibrous valves’  (not all experimental animal models have 4 valves).

Corrected in line 118. Thank you.

Ln112:  Awkward wording:  ‘It has been demonstrated an active mechanical role for the endoderm during this process’:  This sentence needs editing.

Corrected in line 123. Thank you.

Ln122-23:  Replace ‘and pulls’ with ‘to pull’

Corrected in line 134. Thank you.

Ln130:  Replace ‘The particular’ with ‘Specific’  (particular is redundant with the next sentence)

We have rephrased the text (lines 138-160) to explain the concept that mechanotransduction is involved in each stages of cardiac tissue evolution, from the initial specification to final maturation. Thank you for the suggestion.

Ln137:  Delete (‘also’)

Corrected. Thank you.

Ln166:  Delete ‘localized’ (redundant with localize)

Corrected. Thank you.

Ln177:  Change: ‘also tissue mechanics could irreversibly’ to ‘tissue mechanics could also irreversibly’

Corrected in line 193. Thank you.

Ln209:  Change: ‘cell mechanics plays’ to ‘cell mechanics play’

Corrected in line 222. Thank you.

Ln 230:  Replace: ‘cells undergo to endocardial-mesenchymal transition’ with ‘cells undergo an endocardial-mesenchymal transition’

Corrected in line 241. Thank you.

Ln258:  Replace: ‘been in particular observed in’ with ‘been characterized by’

Corrected in line 270. Thank you.

Ln259:  Break up the sentence: ‘ …a study from our group.  This research showed that human VICs, …’

Corrected in line 271. Thank you.

Ln289:  Delete ‘from’:  use ‘rescue the remodeling’

Corrected in line 305. Thank you.

Round 2

Reviewer 1 Report

No further changes needed